# Circulating Microbial Cell-Free DNA in Health and Disease

**DOI:** 10.3390/ijms24033051

**Published:** 2023-02-03

**Authors:** Bernadeta Pietrzak, Iwona Kawacka, Agnieszka Olejnik-Schmidt, Marcin Schmidt

**Affiliations:** Department of Biotechnology and Food Microbiology, Poznan University of Life Sciences, 60-627 Poznan, Poland

**Keywords:** circulating microbial cell-free DNA, non-invasive biomarker, microbial translocation

## Abstract

Human blood contains low biomass of circulating microbial cell-free DNA (cfmDNA) that predominantly originates from bacteria. Numerous studies have detected circulating cfmDNA in patients with infectious and non-infectious diseases, and in healthy individuals. Remarkable differences were found in the microbial composition of healthy subjects and patients compared to cohorts with various diseases or even patients with diversified prognoses, implying that these alterations may be associated with disease development. Although the function of circulating cfmDNA needs to be elucidated (whether it acts as a bystander of dysbiosis or a key player in disease development), several studies have demonstrated its potential as a non-invasive biomarker that may improve diagnosis and treatment efficacy. The origin of circulating cfmDNA is still the subject of much deliberation, but studies have identified members of various microbiome niches, including the gut, oral cavity, airways, and skin. Further studies investigating the origin and function of circulating cfmDNA are needed. Moreover, low-biomass microbiome studies are prone to contamination, therefore stringent negative experimental control reactions and decontamination frameworks are advised in order to detect genuine circulating cfmDNA.

## 1. Introduction

Circulating cell-free DNA (cfDNA) comprises fragments (~160 bp) of double-stranded deoxyribonucleic acid that are present in the blood plasma or serum and other body fluids, and are not encapsulated in cells [1]. It was initially discovered by Mandel et al., 1948, in the blood plasma of healthy subjects [2]. Circulating cfDNA serves as a reservoir of genetic information from all body cells, and has been found to have potential in clinical applications [3]. For instance, it was demonstrated that human-derived cfDNA may be used in tumor diagnosis and monitoring (cell-free tumor DNA, cftDNA) [4], in the prenatal screening of abnormal chromosomal karyotypes (cell-free fetal DNA, cffDNA) [5], in the monitoring of transplant rejection (donor-derived cfDNA, dd-cfDNA) [6], and in the detection of organ dysfunction [7]. Numerous mechanisms of cfDNA release from cells have been proposed, such as apoptosis, necrosis, neutrophil extracellular trap-(NET-)osis, or active secretion via exosomes [8,9]. To date, the process of cfDNA fragmentation has not been fully elucidated, though apoptosis is known to involve nucleases that may contribute to cfDNA generation. Moreover, nucleases are involved in cfDNA clearance from the bloodstream, and the estimated half-life of cfDNA is 16 min to 2 h [9].

Furthermore, growing evidence suggests that blood samples also harbor circulating microbial cfDNA (cfmDNA). A large cohort study investigating circulating cfDNA reported that an average of 0.45% of sequences (reads) did not align with the reference human genome, suggesting that it is of non-human origin [10], which is in agreement with results obtained in another study [11]. The research group revealed the presence of hundreds of novel bacteria and viruses that represented previously unidentified members of the human microbiota (microbiota as a collective of cells and virus particles), as only approximately 1% of non-human reads could be identified in microbiome databases [10]. Over two-thirds of the sequences were bacterial in origin; however, novel phages and anelloviruses were also identified. Similar findings were presented by Tong et al., 2022, who found that the cfmDNA predominantly originated from bacteria (more than 95%), but eukaryotic and viral sequences were also detected [12]. The predominance of anelloviruses in the blood virome of healthy humans was demonstrated [13]. Numerous studies consistently reported that the circulating cell-free microbiome (microbiome as a collective of microbial sequences, representing genetic material of microbiota) was dominated by the Proteobacteria phylum, followed by phyla Actinobacteria, Firmicutes, and Bacteroidetes that were present to a lesser extent [10,14,15]. According to these findings, a hypothesis of core circulating cell-free microbiome existence was stated. Whittle et al., 2019, found that the main members of the circulating cell-free microbiome that were detected in their study have been also identified in previous studies, suggesting the existence of a core microbiome [14]. Xiao et al., 2020, reported that the majority of identified species were found in all study groups, i.e., patients with colorectal neoplasia and healthy subjects, and only a limited number of species were unique in particular groups [11]. In contrast, variations in the microbial composition and the diversity of potential origins of circulating cfmDNA observed in other studies implied that the definition of core circulating cell-free microbiome might be a challenge [16], or that there is no so-called core circulating cell-free microbiome [7]. Furthermore, geographical patterns in the abundance, diversity, and complexity of circulating cell-free microbial profiles, and coexistence networks between species, were also found [11,12]. Taken together, in light of recent studies, this aspect seems to be more complex than it was previously thought and demands further examination.

In this review, we presented findings from recent reports that demonstrated circulating cfmDNA levels and composition in patients and healthy subjects, and the potential origin of cfmDNA in the bloodstream. Moreover, we discussed experimental procedures introduced in various low-biomass microbiome studies to avoid potential environmental contaminations. Further, we summarized potential applications of circulating cfmDNA as a non-invasive biomarker in clinical practice.

## 2. Circulating cfmDNA or Contamination?

Healthy human blood has been considered a sterile environment, and the presence of microorganisms in the blood was found to be associated with life-threatening bloodstream infections (BSIs) [17]. Moreover, the blood environment provides unfavorable conditions for microorganism development [14]. However, microorganisms were detected in the blood of patients with infectious and non-infectious diseases, and in healthy individuals, and a hypothesis developed that these bacteria may reside in the blood mostly in the dormant form [18]. Moreover, the development of molecular techniques, in particular 16S rRNA gene sequencing and shotgun metagenomics, enabled the detection of microbial DNA in health and disease [19]. However, it is worth noting that the detection of microbial DNA is not evidence of viable bacteria presence in the blood. Analysis of microbial DNA extracted from whole blood samples may correspond either to circulating/blood cell-associated microorganisms, or to circulating cfmDNA potentially released from degradated microbial cells (throughout phagocytosis, NETosis, and Membrane Attack Complexes) or actively secreted from microbial cells. Therefore, in this review, we focused only on studies that selected cell-free components of whole blood, such as plasma and serum for DNA extraction, especially with techniques dedicated to cfDNA isolation (omitting cell lysis or disintegration step). Païssé et al., 2016, reported that the blood plasma of healthy donors contained significantly less bacterial DNA than a buffy coat or red blood cells, and its bacterial diversity was significantly decreased as compared to red blood cell fraction. Moreover, differences were found in the taxonomic profiles between particular blood fractions [15]. Results obtained from analysis of samples containing low microbial biomass, such as blood plasma or serum, are highly prone to environmental contamination; therefore contaminant-controlled studies are required to detect genuine cfmDNA in those samples [20]. Firstly, contaminant microbiota may be introduced into the sample during blood collection procedures, and any contamination that would occur at this stage may also affect all downstream procedures. Therefore, in addition to the cleansing of the skin’s surface, it was recommended that the first volume drawn be transferred to a separate tube and that the sample fractions be analyzed independently [14]. Secondly, some procedures to avoid contamination with microorganisms from the laboratory environment were also recommended. For instance, Qiu et al., 2019, reported that a whole range of experimental procedures were conducted in a laminar airflow bench that was illuminated with a UV lamp prior to use [21]. Additionally, Zozaya-Valdés et al., 2021, showed that plasma DNA extraction was conducted in a biosafety cabinet with disinfected and DNA-cleaned equipment, by limited personnel, who were using disposable surgical gowns and gloves [20]. However, the most challenging is the contaminating DNA that was found to be ubiquitous in DNA extraction kits, PCR reagents, and water, and remarkable differences were revealed in the composition of different kits and kit batches [22]. Therefore, it is advised to include negative experimental control reactions that would mirror the whole range of experimental procedures in which plasma or serum would be replaced with molecular biology-grade water or buffer [7,20]. Furthermore, studies have highlighted the importance of an analytical decontamination framework that enables the filtering of amplicon sequence variants with batch-wise abundances and those with a higher prevalence in negative controls, in order to accurately detect circulating cfmDNA [20]. Overall, it was revealed that there is a low (but significantly higher than in negative controls) concentration of circulating cfmDNA in plasma samples, and significant differences were found in the microbial composition of plasma samples compared to negative controls [7,20,23]. Taken together, the evidence favors the idea of cfmDNA existence in blood.

## 3. The Presence of Circulating cfmDNA in Patients and Healthy Individuals

Numerous reports have demonstrated the presence of circulating cfmDNA in patients with various communicable and non-communicable diseases [7,9,12,14,15,18,19,21,22,23,24,25,26,27,28,29,30,31,32,33,34,35,36,37]. Moreover, those studies provided information about circulating cfmDNA in healthy humans, as they mostly included healthy control groups. Recent reports revealed remarkable differences in the concentration and composition of circulating cell-free microbiome between patients and healthy individuals, and the alterations in the microbiome profile appeared to be associated with particular disease development. Overall, elevated levels of circulating cfmDNA were found in patients with inflammatory bowel disease (IBD), Kawasaki disease (KD), human immunodeficiency virus (HIV) [16], cardiovascular diseases (CVD) [25], cystic fibrosis (CF) [27], and pneumonia [38] as compared to healthy individuals. Moreover, circulating cell-free microbiome was characterized by lower diversity in patients with sepsis [7], IBD, KD, HIV [16], gastric cancer (GC) [24], hepatocellular carcinoma (HCC) [28], and melanoma [20] than in healthy individuals. In KD and HIV patients, circulating cell-free microbiome was predominated by Proteobacteria phylum, whereas in IBD patients, the microbiome profile was more complex, and contained mainly Bacteroidetes phylum, followed by Proteobacteria and Firmicutes phyla [16]. There were also 173 genera found that were preferentially abundant in healthy or diseased subjects, and which provided so-called molecular phenotypes of the corresponding diseases. Another study demonstrated increased diversity of circulating cfmDNA and enrichment of Actinobacteria phylum and bacteriophages in CVD patients as compared to healthy individuals [25]. In contrast, no difference in the diversity and composition of circulating cfmDNA at the phylum level was found between patients with type 2 diabetes (T2D) and non-T2D control cohort [21]. However, at the deeper taxonomic level, some variations were found in the relative abundance of relevant taxa between study groups. In particular, a lower abundance of genera *Aquabacterium*, *Xanthomonas*, and *Pseudonocardia*, and a higher abundance of genera *Actinotalea*, *Alishewanella*, *Sediminibacterium*, and *Pseudoclavibacter* was observed among T2D patients as compared to those non-T2D controls. Another key finding of the study was that the circulating cell-free microbiome may act as an etiological factor in T2D development, as it was found that the *Bacteroides* genus carrier was associated with decreased risk for T2D development, whereas the inverse correlation was found in the case of the *Sediminibacterium* genus carrier. Several studies demonstrated that the circulating cell-free microbiome of cancer patients was significantly different from that of healthy individuals [11,24,26,28,29]. In GC patients or healthy subjects, seven microbial taxa were found to be significantly enriched [24]. In particular, genera *Haemophilus*, *Acinetobacter*, *Bacteroides*, and *Haemophilus parainfluenzae* species were enriched in the GC group, while genera *Sphingomonas*, *Comamonas*, and *Pseudomonas stutzeri* species were enriched in the healthy individuals. Another study revealed that the circulating cell-free microbiome was more diverse in early-onset breast cancer (EOBC) patients than in healthy females, and there was a higher abundance of *Pseudomonas* and *Sphingomonas* species in EOBC patients, while in healthy females, most circulating cfmDNA was derived from *Acinetobacter* species [26]. In HCC patients, there was an increased abundance of Proteobacteria phylum, and at the deeper taxonomic level, the abundance of seven bacterial genera—*Pseudomonas*, *Streptococcus*, *Bifidobacterium*, *Staphylococcus*, *Acinetobacter*, *Klebsiella*, and *Trabulsiella*—was significantly different between HCC patients and healthy individuals, among which genus *Staphylococcus* showed the strongest association with HCC [28]. Furthermore, it was demonstrated that differences in the circulating cell-free microbiome profiles enabled distinction between patients with stage III–IV prostate, lung, or skin cancers from healthy individuals, and also cancer-versus-cancer distinction between patients with various cancer types [29]. In contrast, another study found no statistically significant differences in the circulating cell-free microbiome structure between melanoma patients and healthy individuals; however, the genus *Castellaniella* was present only in the healthy cohort [20].

Furthermore, studies investigating the circulating cell-free microbiome in healthy subjects demonstrated several variables that may affect its composition. It was reported that bacterial DNA was present in the serum, plasma, and blood cells of healthy children and adults; however, it was not detected in blood samples from newborn babies [16]. This finding suggests that bacterial DNA potentially originates from microorganisms residing in other body niches, and as the microbiome is immature in newborn babies, it cannot act as a reservoir of bacterial DNA in those subjects. The research group also found that the concentration of circulating cfmDNA in plasma was lower in children than in adults, and in children the concentration was observed to increase with the age, while in adults the concentration was stable. These findings imply that circulating cfmDNA may gradually increase and stabilize with age. Differences were also revealed in the composition of the circulating cell-free microbiome. Briefly, Bacteroidetes phylum (58%) was predominant in children, and Firmicutes phylum (46%) in adults. Furthermore, another study indicated differences in the abundance, diversity, and composition of circulating cell-free microbiomes between distinct regions that could be associated with diet, geography, and environmental factors [12].

Taken together, although there were some exceptions, recent studies have consistently demonstrated the high similarity of circulating cell-free microbiome in healthy and diseased individuals at the phylum level. In contrast, at the deeper taxonomic level, remarkable differences were found either between healthy subjects and patients, or between cohorts of patients with various diseases or even various prognoses. However, the findings presented are preliminary, and further large-cohort studies are required to understand the circulating microbial signatures associated with relevant diseases, in order to improve diagnosis and treatment efficacy. Moreover, it would be valuable if matching data were available for major microbiota, e.g., gut, mouth, vagina, skin, and cancer tissue, to uncover potential sources of the cfmDNA. Additionally, it is worth noting that rigorous control for environmental contaminants are needed to detect genuine cfmDNA, therefore findings from studies that have not included stringent negative experimental control reactions should be taken with caution.

## 4. The Potential Origin of Circulating cfmDNA

Although circulating cfmDNA was detected in various cohorts of patients with communicable and non-communicable diseases, and in healthy individuals (including studies with stringent negative experimental control reactions and decontamination frameworks) [7,11,12,14,16,17,20,23,24,25,26,27,28,29,30,31,32,33,34,35,36,37,38], the presence of core circulating cell-free microbiome in humans is still the subject of much deliberation. The potential origin of circulating cfmDNA is shown in Figure 1.

It is known that pathogens can directly invade the bloodstream or disseminate through the disruption of endothelial and epithelial barriers [27]. The circulating cfmDNA of clinically relevant pathogens was found in patients with BSIs and deep-seated infections, and its load was reduced during antimicrobial treatment, suggesting that microbial translocation may occur periodically during infections [30,31,32,34,35]. The circulating cell-free microbiome could originate from various body localizations, as infections of the abdomen, lung, genitourinary tract, peripancreatic lymph node, heart, brain, sternum, joint, pancreas, chorion, and amnion were identified in those patients [23,30,31,32,33,34,37,38]. It is worth noting that circulating cfmDNA may originate simultaneously from various body sites. In a study conducted on patients with suspected infected pancreatic necrosis (IPN), four cases were found of false-positive results of circulating cfmDNA sequencing as compared to the results of peripancreatic drains culture, and the identified pathogens were associated with cholecystitis or ventilator-associated pneumonia [32]. However, active infections may not be the only source of microorganisms. For instance, it was reported that elevated concentrations of total and pathogen-derived circulating cfmDNA, and higher diversity of identified organisms in COVID-19 non-survivors imply that the circulating cell-free microbiome could originate either from secondary pneumonia or from respiratory tract microbiota translocation due to COVID-19-associated lung injury [33]. Moreover, the presence of the circulating cfmDNA of respiratory pathogens was revealed in CF patients, and the research group suggested that several factors, such as extensive lung vascularization, pathogen expression of invasive functions, and lung epithelium injury induced by inflammation, could raise the possibility of microbial translocation from the lungs to the bloodstream of CF patients during chronic infection [27]. Moreover, it was reported that postoperative patients exhibited significantly elevated levels of 16S rRNA gene copies in the blood; however, non-surgical procedures, such as catheters, needles, and hemodialysis could also be a source of microorganisms that may invade the bloodstream [39,40,41]. Additionally, it was reported that toothbrushing increased the prevalence of bacteremia in patients with plaque accumulation and gingival inflammation, and significant compositional differences were found in the blood microbiome between periodontally healthy and periodontally diseased cohorts [42]. Therefore, a study investigating the blood microbiome composition of healthy individuals included a criterion that donors who had undergone dental treatment, surgical intervention, or body piercing had to wait a defined time interval after the procedure to be recruited to the study [15]. There is also a hypothesis that the presence of microorganisms or their parts, e.g., microbial nucleic acids in the bloodstream, may be a consequence of their translocation from other microbiome niches within the body, such as gastrointestinal; respiratory and genitourinary tract; oral cavity; and skin [14,15]. There is also a scarcity of reports exploring the mechanisms of microorganism or cfmDNA entry into circulation; however, several hypotheses have been proposed, such as entry via micro-fold cells, dendritic cells, or a dysfunctional epithelial barrier [18,43]. A recent study supported this hypothesis by demonstrating the potential role of the paracellular permeability of epithelial cell layers in bacterial translocation into the bloodstream [44]. A significant association was found between the concentration of 16S rRNA gene copies in the whole blood samples and serum zonulin levels, which is known as a marker of intestinal permeability, suggesting that the bacterial DNA may originate from the gastrointestinal tract. However, zonulin was also found to be a regulator of intercellular tight junctions in the lungs, and bacteria that represent dominant members of the lung microbiota were identified, indicating another possible origin of bacterial DNA in the bloodstream [45,46]. Moreover, Whittle et al., 2019, performed an in silico comparison of the data from healthy and asthmatic subjects with the Human Microbiome Project (HMP) data, and found that the plasma microbiome of the cohort most likely originated from the oral or skin communities rather than from microorganisms that colonize the gastrointestinal tract [14]. Moreover, the *Achromobacter* genus was the most abundant in both groups and it is known to be a respiratory pathogen [47], suggesting that the blood microbiome could also originate from the respiratory tract. However, the research group also detected viable microorganisms in 80% of plasma samples and suggested that positive culture could result from venepuncture contamination of the blood sample with skin bacteria, or that these bacteria were present in the blood in a dormant state and were revived during culture procedure [14]. Zhao et al., 2020, found that genera identified in the plasma samples of healthy subjects represented members of airway, stool, oral, or skin environments (according to the HMP database), whereas in patients with IBD, KD or HIV the potential origin was less diversified [16]. In KD patients genera mostly originated from the skin, and in HIV patients it originated from airways, skin, and other body regions. In contrast, in IBD patients the gut microbiome was found to be the main source of the circulating cell-free microbiome, and it was suggested that intestinal bacteria could enter into the bloodstream due to the impaired mucosal barrier integrity that is commonly found in IBD patients [48]. Several observations supported this hypothesis: firstly, IBD patients had remarkably increased concentrations of circulating cfmDNA as compared not only to healthy subjects (~100-fold), but also compared to patients with KD (~10-fold) or HIV (~8-fold) [16]; secondly, there was a higher abundance of Bacteroidetes and Firmicutes phyla in those patients as compared to other study groups, and the most abundant genera were reported to be associated with dysbiosis and obesity; thirdly, differences were found in bacterial abundance between pre- and post-treatment IBD patients, implying that circulating cfmDNA might indicate changes in the gut microbiota and/or changes in intestinal barrier permeability. Consistent results were presented in another study demonstrating that the majority of circulating cfmDNA in patients with colorectal neoplasia and in healthy subjects originated from the gastrointestinal tract, oral tract, and skin [11]. Moreover, a moderate positive correlation was found between fold changes of the overlapped fecal and circulating cfmDNA, suggesting that alterations in the circulating cell-free microbiome profile may reflect the dysregulated gastrointestinal microbiome and inflammatory status of the gut mucosa.

## 5. Potential of Circulating cfmDNA in Clinical Applications

As mentioned above, alterations in the circulating cell-free microbiome were reported in patients with various communicable and non-communicable diseases. Although the function of circulating cfmDNA remains unclear (whether it acts as a bystander of dysbiosis or a key player in disease development), its potential as a non-invasive biomarker of particular diseases has been recently investigated. The major findings of selected studies demonstrating the potential of circulating cfmDNA in clinical applications are summarized in Table 1.

A growing number of studies have demonstrated that circulating cfmDNA sequencing may serve as a sensitive and specific assay for accurate diagnosis of BSIs and sepsis [7,17,23,30,35]. Conventional blood cultures for the detection of BSIs are limited by long turnaround time, a limited spectrum of pathogens (the presence of non-culturable pathogens), dependence on antimicrobial treatment, and decreased sensitivity [7,35,49]. Moreover, the common treatment approach involves the use of empiric and broad-spectrum antibiotics that may not affect specific pathogens and/or may cause harmful adverse events [49]. Therefore, novel diagnostic methods are required to accurately identify pathogenic microorganisms and select adequate treatment strategies. The advantage of circulating cfmDNA sequencing over conventional blood cultures was revealed in several recent studies on BSI and septic patients [7,17,23,30,35]. In contrast to conventional blood cultures, the method was characterized by a higher positivity rate for pathogen identification, and the circulating cfmDNA of the causative pathogen was detectable for significantly longer [30], which is in agreement with another study [35]. Moreover, the vast majority of the results were assessed as plausible and would have led to a change to a more adequate antimicrobial therapy in more than half of the cases [30]. Notably, a retrospective analysis showed that in patients who were treated adequately (according to cfmDNA sequencing results), higher 28- and 90-day survival rates were observed, together with an overall reduction in the use of antimicrobials. Another key finding was that the duration of the circulating pathogen’s cfDNA was associated with an increased risk for metastatic infection [35]. Moreover, circulating cfmDNA sequencing enabled the identification of pathogens in the days before the onset of infection in immunocompromised pediatric patients with relapsed or refractory cancer, who are at high risk of BSIs [17]. A recent study indicated that integration of both circulating microbial and human cfDNA information into a machine learning model improved performance in sepsis diagnosis and mortality prediction as compared to models with any individual parameters [7]. Furthermore, the clinically relevant pathogens in patients with invasive fungal infections (IFIs) could also be non-invasively detected by the sequencing of circulating cfmDNA [23]. Current diagnostic technology is based on carrying out an invasive biopsy, which generates high costs and is associated with high morbidity. Moreover, available non-invasive biomarkers, such as *Aspergillus* galactomannan and beta-D-glucan tests, have limitations in the detection of a wide range of non-*Aspergillus* molds. The research group demonstrated that the sequencing of circulating cfmDNA detected both *Aspergillus* and non-*Aspergillus* molds in patients with proven IFIs. In one case, the sequencing assay enabled the differentiation of the *Aspergillus lentulus* from the *Aspergillus fumigatus* species complex, which could have important clinical consequences. Briefly, although these microorganisms are morphologically identical and share 91% of sequence identity, it was reported that *A. lentulus* showed decreased susceptibility to many azoles [50]. Moreover, it was estimated that the utility of the sequencing assay in the identification of pathogens in IFI patients was associated with cost savings [51].

The sequencing of circulating cfmDNA could also be used for the accurate detection of causative pathogens in other infections [31,32,33,34,36,37]. A recent study showed that the sequencing assay may provide an alternative to conventional cultures and bronchoscopy for the comprehensive identification of secondary pneumonia in COVID-19 patients, and may enable better antibiotic stewardship in those patients [33]. Furthermore, circulating cfmDNA sequencing enabled the accurate identification of causative pathogens in patients with infective endocarditis [34]. Similarly, as in BSI and septic cases [30,35], the duration of positivity from antibiotic treatment initiation was significantly longer for circulating cfmDNA than for conventional blood cultures. It was therefore suggested that the sequencing assay also offers a novel method to estimate the burden of infection in patients and control response to treatment [34]. Comparable findings were presented in another study [31]. The use of circulating cfmDNA sequencing in addition to tissue cultures increased the number of cases in which pathogens causing local periprosthetic joint infections were accurately identified. Additionally, the sequencing assay improved the time to species identification as compared to standard-of-care methods and could be utilized to control infection clearance during the treatment. It was also reported that the circulating cfmDNA of pathogens relevant to chorioamnionitis, neonatal sepsis, and intra-amniotic infections were significantly increased in the maternal plasma of women with clinical or histological chorioamnionitis as compared to the control cohort without chorioamnionitis [37]. Additionally, species *Streptococcus mitis*, *Ureaplasma parvum*, and *Mycoplasma hominis* significantly correlated between matched maternal and umbilical cord plasma samples, implying that these microorganisms were causative pathogens. These findings suggested the potential of circulating cfmDNA sequencing in the non-invasive detection of perinatal infections, which would enable targeted therapy commencement and reduction of adverse outcomes in mothers and neonates. Furthermore, there was a higher positivity rate for cfmDNA-sequencing-based identification of pathogens in patients with suspected IPN [32]. More bacterial and fungal species were detected using cfmDNA sequencing than using conventional blood culture, and the sequencing results were consistent with peripancreatic drains cultures. As previously reported, circulating cfmDNA sequencing also improved the time to pathogen identification and the results were less affected by antibiotic treatment. Additionally, patients with positive sequencing results had new-onset septic shock significantly more frequently and needed percutaneous catheter drainage and surgical intervention. Circulating microbial cfDNA sequencing may also serve as a sensitive and specific test in the diagnosis and treatment optimization in patients with febrile neutropenia [36]. The detected pathogens were assessed as a plausible cause of febrile neutropenia in every case, and polymicrobial infections were revealed in 61% of positive samples, mostly associated with gastrointestinal syndromes. Moreover, it was shown that circulating cfmDNA sequencing could enable more timely and appropriate treatment.

Numerous studies have demonstrated the association between the gut microbiome and cancer development, in particular gastrointestinal and hepatobiliary malignancies, and its composition and diversity is correlated with the clinical outcomes of immune checkpoint inhibitor therapy in various cohorts of cancer patients [52,53]. Moreover, growing evidence suggests that the circulating cell-free microbiome may become a novel biomarker for cancer diagnosis [54]. Huang et al., 2018, in a study on a small cohort of EOBC patients and healthy females, indicated that circulating cfmDNA has potential as a prognostic indicator [26]. Briefly, the patient with a circulating cell-free microbiome profile similar to that of healthy controls had durable disease-free survival, in contrast to those with a more diverse profile who had short disease-free survival. Moreover, circulating cfmDNA sequencing was presented as a potential tool for early diagnosis and characterization of GC [24]. The combination of six microbial biomarkers achieved a high classification power for GC and healthy individuals. Moreover, fifteen genera and species correlated significantly with clinical parameters, such as tumor-node-metastasis stage, lymphatic metastasis, tumor diameter, and invasion depth in GC. Additionally, differences were found in the circulating cell-free microbiome structure between patients with GC-lymphatic metastasis (LM) and non-LM, and two genera *Enterococcus* and *Bacteroides* were enriched in GC-LM. Another study showed a five-genera microbiome signature model that accurately distinguished HCC from healthy controls [28]. Furthermore, Xiao et al., 2020, identified 28 species that enabled accurate differentiation of patients with colorectal cancer/colorectal adenoma from healthy subjects [11]. These findings suggest that the circulating cell-free microbiome may also become a non-invasive biomarker for screening and early diagnosis of HCC or colorectal neoplasia [11,28].

Taken together, recent findings suggest that circulating cfmDNA sequencing provides a valuable non-invasive method for the accurate identification of clinically relevant pathogens (independently of antimicrobial treatment commencement) and the prediction of the outcome in patients. Therefore, cfmDNA sequencing may support clinicians in the selection of an adequate treatment strategy, which potentially could reduce broad-spectrum antimicrobial use, antimicrobial-associated toxicity, and infection-related mortality in patients. Moreover, the potential of cfmDNA sequencing in oncology has been shown. However, despite the advantages of circulating cfmDNA detection through the use of sequencing technology, there are also some limitations to the widespread use of the assay in diagnostics. Firstly, as plasma cfDNA harbors a remarkably higher number of human-derived cfDNA fragments than those derived from microorganisms, sample processing methods for human cfDNA depletion alongside postprocessing bioinformatic removal are inevitable, making the analysis tedious and costly. Additionally, background human cfDNA affects the sensitivity of the assay, and due to its presence greater sequencing depth for microorganism identification is required. Finally, it is worth noting that the detection of cfmDNA may indicate a clinically relevant pathogen contributing to the development of particular diseases; microorganisms originating from other active infections; human microbiota; or environmental contamination, which may affect the diagnostic performance of plasma cfmDNA sequencing technology [55,56].

## 6. Conclusions and Future Directions

According to recent studies, mounting evidence suggests the presence of circulating cfmDNA in patients and healthy subjects, and altered circulating cell-free microbiome composition is associated with particular diseases. The use of high-throughput sequencing technologies enabled the detection of numerous previously uncharacterized microorganisms that should be extensively explored to broaden our knowledge of complex human microbiomes and their role in health and diseases. Firstly, cfDNA is considered a reservoir of genetic information from all (human and microbial) cells; therefore, its analysis presents a new approach that would enable the microbiome characterization of inaccessible body niches. Secondly, among those uncharacterized microorganisms, there may be causative pathogens of diseases with currently unknown etiology. Thirdly, circulating cfmDNA may become a valuable non-invasive biomarker for the prediction or early detection of particular diseases, which may improve the selection of adequate therapy and reduce mortality rates. However, further large cohort studies are warranted to better understand the potential of circulating cell-free microbiome as a biomarker in various diseases. Moreover, several aspects are still the subjects of much deliberation—such as the presence of the so-called core circulating cell-free microbiome, the origin of circulating cfmDNA, and its function—and demand further examination. Subsequent analysis of a few samples reflecting various human microbiomes, such as blood, stool, and saliva samples, from the same group of participants, could reduce the impact of individual heterogeneity on the results, and enable exploration of circulating cfmDNA sources. Moreover, multi-omic studies, including transcriptome, proteome, and metabolome, could improve understanding of the biological function of an altered circulating cell-free microbiome in the development and progression of various diseases.

## Figures and Tables

**Figure 1 ijms-24-03051-f001:**
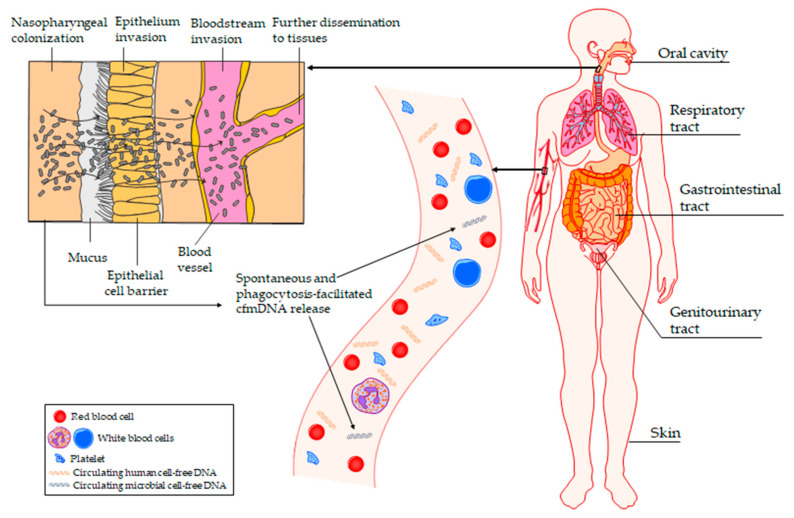
Microorganisms colonizing various body niches, including gastrointestinal, respiratory, and genitourinary tract, oral cavity, and skin, may translocate into the bloodstream through impaired epithelial barrier, as a consequence of toothbrushing, dental treatment, medical procedures, skin injuries, and body piercing.

**Table 1 ijms-24-03051-t001:** Potential application of plasma cfmDNA sequencing in clinical practice.

Clinical Context	Major Findings	References
Septic shock	>70% of the positivity rate for NGS-based pathogen identification over the whole study period. 96% of positive NGS results for acute sepsis time points were plausible. NGS results would have led to a change to a more adequate therapy in 53% of cases.	[30]
Relapsed or refractory pediatric cancers	75% and 80% of predictive sensitivity of NGS for all BSIs and bacterial BSIs, respectively, in the 3 days before the onset of infection. 82% and 91% of the specificity of NGS, for any bacterial or fungal organism and any common BSI pathogen, respectively.	[17]
BSIs	89.3% and 74.3% of the NGS sensitivity and specificity, respectively. NGS identified causative pathogens for a significantly longer interval than conventional blood cultures (median 15 days vs. 2 days; *p* < 0.0001). The odds of metastatic infection significantly increased with each additional day of circulating cfmDNA detection (odds ratio, 2.89; *p* = 0.0011).	[35]
Sepsis	NGS reached the sensitivity and specificity of 0.952 and 1, respectively, for the identification of bacterial infection, and allowed for the simultaneous detection of viral pathogens. NGS revealed differences in the composition of circulating cfmDNA between septic and non-septic patients and between survivors and non-survivors by 28-day mortality. Improved performance was achieved in identifying sepsis and the prediction of clinical outcomes for ICU patients with AUC of 0.992 and 0.802, respectively, by integrating the information from circulating human and microbial cfDNA into a machine learning model.	[7]
Infective endocarditis	NGS achieved a sensitivity of 87%. NGS identified causative pathogens for a significantly longer interval than conventional blood cultures (median duration of positivity from antibiotic treatment initiation was 38.1 days vs. 3.7 days; *p* = 0.02771). The level of cfmDNA significantly decreased after surgical source control.	[34]
Periprosthetic joint infections	NGS identified causative pathogens in 57% of cases, increasing pathogen detection to 94% (as an adjunct to tissue cultures).NGS improved the time-to-speciation (the median time was 3 days less than standard-of-care methods).After treatment, NGS did not detect circulating cfmDNA of the infectious pathogen or there were reduced levels of circulating cfmDNA.	[31]
Infected pancreatic necrosis	The positivity rate of NGS was 54.55%. 83.33% of NGS-positive cases were consistent with the culture results of infected pancreatic necrosis drains. The PPA and NPA of NGS were 80.0% and 89.47%, respectively. Patients in the NGS positive group had new-onset septic shock significantly more frequently (12 (50.0%) vs. 4 (20.0%), *p* = 0.039) than those in the negative group.	[32]
Febrile neutropenia	The PPA and NPA of NGS were 90% and 31%, respectively. NGS sensitivity and specificity were 85% and 100%, respectively. NGS improved the time to diagnosis. NGS results would have led to a change to a more adequate therapy in 47% of cases.	[36]

Next-generation sequencing (NGS); bloodstream infection (BSI); intensive care unit (ICU); area under the curve (AUC); positive percent agreement (PPA); negative percent agreement (NPA).

## Data Availability

Not applicable.

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
