# Peer review of "Circulating Microbial Cell-Free DNA in Health and Disease"

_ijms, 2023, doi:10.3390/ijms24033051_

Round 1

Reviewer 1 Report

The review is clearly written and covers most important aspects of cfmDNA in a clinical context.

Here below are my suggestions:

The title should better reflect the content of the review, for example “circulating microbial cell-free DNA in health and disease”, because more “fundamental” aspects (the origin and mechanisms of mcfDNA release and fragmentation, size and stability) have not been dealt with. Nevertheless, the authors can add this information in a short text section (a couple of lines). The current limitations for a massive use of cfDNA NGS in diagnostics could also be mentioned in the ms.

Line 49. It is not clear what does “exogenous” refer to.

Line 88. “immune degradation” could be reworded for clarity.

Line 148-156. The results of this study should be taken with precaution. The bacteria found to be differentially abundant between the groups correspond to environmental organisms, most of them had median values of zero and the subtraction of reagent contaminants has not been clearly explained.

Likewise, bacterial genome loads of 10^6 to 10^7 per mL blood found in the study of Païssé (reference 13) reveal serious technical issues and inappropriate data interpretation in the original study.

Therefore, if the authors prefer not to critically evaluate these findings, they could point in a more general to the need for more rigorous control for reagent (and other, e.g. skin) contaminants  than it was the case in many studies conducted so far. This could be stated, for example, in the last paragraph of section “3”.

Line 211. Actually, it “is shown” (not “was shown”).

Figure 1 could be improved by adding cfDNA fragments of human origin (which are dominant and could be represented by a different color) and few bacterial fragments. The site of origin of mcfDNA could be listed in the symbol legend.

Line 291-292. Could it also be due to changed gut permeability?

Line 315. It is unclear what is meant by “lack of specificities”.

Line 327. Perhaps indicate “cfm DNA sequencing results”.

Line 346. “sequence identity” instead of “sequence homology”.

Line 425. Perhaps indicate “all (human and microbial) cells”.

Line 450. Please check/correct “The was designed”.

References. The journal title is missing in some references e.g. #7, #41 (and possibly other) and the ms title is not specified e.g. #50.

Table 1. I suggest to (1) change the title to “Potential application of plasma cfmDNA sequencing in clinical practice”, (2) replace “Samples” by “Clinical context” in the table heather and (3) remove throughout “Plasma samples from patients with” (so to read “Septic shock” , “Relapsed or refractory pediatric cancers…”

Reviewer 2 Report

The authors describe a very interesting and innovative topic. Recently, there has been interest in circulating microbial cell-free DNA and its potential role in the diagnosis and prognosis of many diseases. But for now, this should be approached with great caution, especially since NGS-based tests are not standardized for full clinical microbiological diagnosis.

Overall, the manuscript is very interesting and sparingly written, though minimal corrections are needed:

-lines 147-155: now the term "diabetes mellitus" is no longer used, but "diabetes", so the abbreviation should also be modified - T2D (type 2 diabetes)

-I have doubts about Figure 1. It is aesthetic, but it does not bring anything new and can be successfully replaced by text. Perhaps enriching it with the addition of diagrams of individual niches (oral cavity, intestines...) from which cfm DNA originates would make more sense? It could also be a graphical abstract of the manuscript.
